# A Retrospective Evaluation of the Steroid-Sparing Effect of Oral Modified Ciclosporin for Treatment of Canine Pemphigus Foliaceus

**DOI:** 10.3390/vetsci9040153

**Published:** 2022-03-23

**Authors:** Eric Chong, Michaela Austel, Frane Banovic

**Affiliations:** College of Veterinary Medicine, University of Georgia, Athens, GA 30602, USA; ejchong89@gmail.com (E.C.); maustel@uga.edu (M.A.)

**Keywords:** canine, pemphigus foliaceus, ciclosporin

## Abstract

The efficacy of ciclosporin as an adjuvant immunosuppressant administered with glucocorticoids (GCs) for induction of canine PF remission is unknown. This study is a retrospective review of medical records from 2015 to 2020 to evaluate the therapeutic outcomes of 11 PF dogs treated with oral modified ciclosporin and GCs. Concurrent GCs were given with ciclosporin to all PF dogs. Nine dogs (9/11) achieved complete remission (CR); five dogs received ciclosporin at a mean dose of 6.2 mg/kg/day; and four dogs received a combination of ciclosporin and ketoconazole at a mean dose of 3 mg/kg/day, respectively. Two dogs (2/11) showed only 25% or poor response, with the development of new PF lesions during treatment. The mean duration of ciclosporin therapy for nine dogs to achieve CR was 65 days (median 57 days, range 24–119 days). Slow tapering of oral GCs while continuing ciclosporin at the same dose and frequency in nine dogs with CR led to recurrence of PF lesions in four dogs, whereas, in five dogs, oral glucocorticoids were discontinued without a PF flare. Oral modified ciclosporin combined with GCs achieved CR in 9 out of 11 PF dogs during the induction phase in this study.

## 1. Introduction

Pemphigus foliaceus (PF) is one of the most common canine cutaneous autoimmune diseases associated with a production of pathogenic autoantibodies that predominantly target the epidermal adhesion proteins desmocollin-1 (Dsc-1) and desmoglein-1 (Dsg-1) [1]. Therapeutic management of canine PF is challenging and commonly requires immunosuppressive medications to achieve long-term clinical remission [1]. Although glucocorticoids (GCs) are considered the mainstay of therapy for canine PF, oral high-dose GC monotherapy (prednisone or prednisolone given at ≥2 mg/kg/day) induced complete remission (CR) in only 15% of dogs with PF within the first three months of treatment [2]. In addition, severe adverse effects of systemic GCs are frequently observed during PF treatment, and some dogs were humanely euthanized due to intolerable adverse effects [2]. Therefore, steroid-sparing adjuvants (e.g., azathioprine, ciclosporin, mycophenolate mofetil, leflunomide) have been emphasized to help induce earlier clinical remission and PF disease control, with each of the steroid-sparing adjuvants having its own specific advantages and disadvantages [3]. However, it remains challenging for clinicians to select the optimal steroid-sparing adjuvant due to the lack of prospective comparative clinical trials with canine PF patients [4].

Oral modified ciclosporin has been used in dogs with various autoimmune skin diseases at dosages ranging from 5 to 10 mg/kg/day [5]. In a prospective pilot study, monotherapy with oral modified ciclosporin given at 5–10 mg/kg/day for three months failed to induce clinical improvement in five dogs with PF [6]. In contrast, oral ciclosporin at 8 mg/kg/day given as monotherapy induced complete remission in a single dog with PF [7], challenging the results from the initial failed monotherapy ciclosporin study [5]. Therefore, it is currently unknown whether ciclosporin can be beneficial for therapy of canine PF as a monotherapy or as an adjunct steroid-sparing agent.

The objective of this retrospective study is to expand the knowledge regarding the therapeutic validity of oral modified ciclosporin and GC combination therapy for the induction of remission and maintenance therapy in canine PF patients.

## 2. Materials and Methods

### 2.1. Review of Medical Records

For this study, we retrospectively analyzed online medical records from the Veterinary Teaching Hospital’s record system from January 2015 to August 2021. An initial computerized search was performed using the keywords “canine”, “pemphigus”, and “pemphigus foliaceus” in the diagnosis section to identify all potential canine PF cases. After identification of cases, all PF patients identified were then analyzed in detail for inclusion and exclusion criteria. Ethical consideration and inclusion were not required, since this study is a retrospective evaluation, and there was no randomization of client-owned animals.

### 2.2. Inclusion and Exclusion Criteria

All dogs diagnosed with PF and treated with oral modified ciclosporin were included in the study. There was no restriction regarding oral modified ciclosporin drug dosages, frequency of administration, and duration of treatment. Pemphigus foliaceus was diagnosed in all dogs based on previously published criteria regarding history, clinical features (e.g., erosions and crusts, with or without pustules), and microscopic demonstration of acantholytic keratinocytes with no clinical or microscopic evidence of skin infection (e.g., intracellular cocci in neutrophils, dermatophytosis) [2]. Direct (skin-fixed autoantibodies) and indirect immunofluorescence (circulating anti-keratinocyte autoantibodies) were not required for the diagnosis of PF in included dogs.

We excluded from this case series dogs that either: (1) had incomplete medical records, (2) were lost to follow-up after oral modified ciclosporin treatment and, therefore, the treatment response could not be assessed in detail, (3) had evidence of intracellular bacteria upon skin cytology, (4) had histopathology showing interface dermatitis [2], and (5) had additional immunosuppressive agent (e.g., azathioprine, mycophenolate mofetil, leflunomide) in conjunction with ciclosporin and GCs.

### 2.3. Medical Record Data Analysis

Medical records of included dogs were examined, and the following information was obtained: (1) patient signalment (age, breed, sex, and weight), (2) cytology and/or histopathology findings, (3) age at PF onset, (4) immunosuppressive therapies prior to oral modified ciclosporin treatment, (5) complete blood cell count, chemistry panel, and urinalysis before and after treatment with oral modified ciclosporin when available, (6) initial dose of oral modified ciclosporin for remission induction and a dose for remission maintenance, (7) concurrent use of GC treatments (e.g., type of GC, dosage, frequency, etc.), and (8) adverse effects during oral modified ciclosporin administration.

### 2.4. Clinical Treatment Outcome Assessment

The response to oral modified ciclosporin treatment was assessed during the induction of PF remission (i.e., induction phase) and maintenance of PF remission (i.e., maintenance phase) [8,9]. The response to treatment during the induction phase was evaluated as either complete remission (CR), partial remission (PR), or poor response (POR) [2,10,11]. Full clinical resolution of all skin lesions associated with active PF disease was defined as CR [2,11,12]. Any body areas containing lifted crusts with underlying healed skin or body areas with healing skin lesions void of signs of active inflammation, with or without alopecia, were not considered affected by active PF [4,11]. Partial remission was defined as ≥50% resolution of extent and severity of PF lesions compared to findings upon the patient’s initial visit [4]. Poor response was defined as less than 50% resolution of PF lesions at the end of oral modified ciclosporin treatment compared to the findings upon the patient’s initial visit [11]. During the maintenance phase, dogs were evaluated to check whether PF remission could be maintained with ongoing administration of oral modified ciclosporin while slowly tapering the dose and/or frequency of GCs.

### 2.5. Data Analysis

All data obtained from the patients’ medical records were analyzed descriptively.

## 3. Results

### 3.1. Patient Signalments

The retrospective analysis of case records yielded 22 PF cases that were treated with ciclosporin; however, only 11 canine PF patients fulfilled the inclusion criteria. Six canine PF patients included in this study were initially treated with oral mycophenolate mofetil and previously published in the retrospective study evaluating oral MMF efficacy in canine PF [4].

Among 11 canine PF patients included in this study, the following breeds were represented: Labrador retrievers (three), Maltese (two), mixed-breed dogs (two), Shih tzu (one), Boston terrier (one), Beagle (one), and English bulldog (one). There were six males (four neutered and two intact) and five spayed females. The mean age of canine PF symptom onset in this study was 6.3 years (range 1–10 years, median 6 years). Ten dogs (10/11) with PF showed a classic “facial” lesion distribution pattern along with other affected body parts, including foot pads, limbs, lateral and ventral sides of the thorax and abdomen, axillae, and rump. A “truncal phenotype” of PF without classic facial lesions was observed in a single dog (1/11) in this study.

### 3.2. Immunosuppressive Treatments Prior to Oral Modified Ciclosporin Therapy

Three out of eleven dogs did not receive previous immunosuppressive therapy, while eight out of eleven dogs received previous immunosuppressive treatments, including oral GCs, doxycycline and niacinamide combination therapy, mycophenolate mofetil, and azathioprine.

### 3.3. Drug Dosages and Response to Therapy during the Induction Phase

The details of oral modified ciclosporin and GC dosages administered and treatment outcomes for all PF dogs are summarized in Table 1.

Nine dogs (Dogs 1–9) achieved CR (Figure 1). Of these nine dogs, five dogs received oral modified ciclosporin (Atopica; Elanco, Greenfield, IN, USA) at a mean dose of 6.2 mg/kg/day (range 2.2–9.6 mg/kg/day), and four dogs received a combination of ciclosporin (Atopica; Elanco, Greenfield, IN, USA) and ketoconazole at a mean dose of 3 mg/kg/day (range 2.4–3.5 mg/kg/day) and 3.6 mg/kg/day (range 2.9–5.2 mg/kg/day), respectively. Ketoconazole was provided to increase ciclosporin blood concentrations [13,14].

Concurrent GCs were given with ciclosporin to all nine CR dogs (Dogs 1–9) during the induction phase. The mean dose of oral prednisone, prednisolone, and dexamethasone was 1.5 mg/kg/day (range 0.8–2.5 mg/kg/day, 6/9 dogs), 1.7 mg/kg/day (range 1.5–1.9 mg/kg/day, 2/9 dogs), and 0.1 mg/kg/day (1/9 dogs), respectively. Topical GCs (mometasone furoate 0.1% cream, twice daily) were combined with other systemic immunosuppressive treatments in 5 out of 11 dogs.

Two dogs (Dogs 10–11) received ciclosporin (Atopica; Elanco, Greenfield, IN, USA) at 5.3 mg/kg/day (5–5.5 mg/kg/day) and showed POR with the development of new PF lesions during treatment. These two dogs received prednisone at a mean dose of 1.4 mg/kg/day (range 1–1.7 mg/kg/day) concurrently. Dog 10′s prednisone was switched to dexamethasone at 0.1 mg/kg/day due to a lack of response

### 3.4. Duration of Therapy

The average duration of oral modified ciclosporin therapy in all 11 dogs was 504 days (median 344 days, range 144–1510 days). The mean duration of therapy with GCs and ciclosporin or ciclosporin/ketoconazole in nine dogs with CR was 65 days (median 57 days, range 24–119 days). The mean duration of oral modified ciclosporin therapy for two dogs (Dogs 10–11) with POR was 248 days (median 248 days, range 159–337 days).

### 3.5. Adverse Effects

The observed adverse effects during therapy with GCs and oral modified ciclosporin included diarrhea (Dog 3, 5, 9), inappetence (Dog 4 and 11), vomiting (Dog 7), and gingival hyperplasia (Dog 8). None of the patients discontinued oral modified ciclosporin due to adverse effects.

Baseline blood work before and after initiation of ciclosporin and GC treatment was available for a single dog (Dog 8). Previous treatment for Dog 8 included long-term treatment with prednisone and azathioprine. Before the initiation of cyclosporine/GC therapy, leukocytosis (18.54 × 103, reference range 5.05–16.76 × 10,310/µL) with neutrophilia (13.94 × 103, reference range 2.95–11.64 × 103/µL) and monocytosis (1.49 × 103, reference range 0.16–1.12 × 103/µL) were observed, as well as an increase in alkaline phosphatase activity (1118 U/L, reference range 23–212 U/L) and alanine aminotransferase activity (743 U/L, reference range 10–125 U/L). Blood work obtained after 3 months of treatment with ciclosporin and occasional systemic/topical GCs showed thrombocytosis (483 × 103 µL, reference range 226–424 × 103 µL) and hyperglycemia (133 mg/dL, reference range 76–120 mg/dL).

### 3.6. Maintenance Therapy

Of the nine PF dogs that achieved CR during the induction phase (Dogs 1–9), five PF dogs (Dogs 1, 4, 6, 8, and 9) did not show a flare of PF lesions after a slow taper and subsequent discontinuation of systemic and/or topical GCs during the maintenance phase. After the discontinuation of GCs, tapered doses of ciclosporin (Atopica; Elanco, Greenfield, IN, USA) and ciclosporin/ketoconazole were utilized for the PF maintenance of Dogs 1 and 4 and Dogs 6, 8, and 9, respectively. Dogs 4′s Atopica was switched to a human generic oral modified ciclosporin (TEVA pharmaceuticals; Parsippany, NJ, USA) at the same dose without a PF flare after being on Atopica for 958 days.

Of the nine PF dogs that achieved CR during the induction phase (Dogs 1–9), four dogs (Dogs 2, 3, 5, and 7) showed a PF flare after slow taper and/or discontinuation oral GCs during the maintenance phase. These four dogs (Dogs 2, 3, 5, and 7) required occasional intermittent oral and topical GCs to maintain the success of PF treatment.

## 4. Discussion

The results from this retrospective study demonstrate the steroid-sparing effect of oral modified ciclosporin in managing canine PF. In this study, oral ciclosporin (mean daily dose 6.2 mg/kg/day) and ciclosporin/ketoconazole (mean daily dose 3 and 3.6 mg/kg/day, respectively) in conjunction with oral/topical GCs achieved CR in 9 out of 11 dogs after a median treatment time of 57 days without significant adverse effects. Tapered oral modified ciclosporin from these nine CR dogs successfully maintained PF disease in five dogs. In contrast, four dogs required occasional intermittent oral and/or topical GCs to control PF.

Several canine autoimmune diseases (e.g., immune-mediated hemolytic anemia and thrombocytopenia, cutaneous lupus erythematosus) have been successfully treated with oral modified ciclosporin at dosages from 5 to 16 mg/kg/day [5,15]. Ciclosporin, a calcineurin inhibitor, blocks T-cell infiltration, activation, and the subsequent release of inflammatory cytokines interleukin (IL)-2, IL-4, interferon (IFN)-γ, and tumor necrosis factor (TNF)-α [13,14]. The exact mechanism by which ciclosporin exerts its immunosuppressive effects in pemphigus is not completely understood. Ciclosporin likely exerts the therapeutic effect in antibody-driven autoimmunity (e.g., pemphigus) by controlling B-cell activation trough inhibition of reactive T helper cells interaction with naive B cells [16,17]. Moreover, ciclosporin reduces matrix metalloproteinase-9 expression and blocks both c-Jun *N*-terminal kinase (JNK) and the p38 signaling pathways that are involved in the pathogenesis of pemphigus [17]. In contrast to the initial unsuccessful results from a pilot study of oral ciclosporin monotherapy (5–10 mg/kg/day) in canine PF [6], our study shows that oral ciclosporin (mean dosage 6.2 mg/kg/day) in conjunction with GCs is effective for initial and long-term management of canine PF.

Oral combination therapy of ciclosporin and ketoconazole has been previously utilized for the treatment of perianal fistulas [18] and cutaneous lupus erythematosus [19] in dogs. Ketoconazole is given concurrently with ciclosporin to increase serum ciclosporin levels, which likely occurs via inhibiting intestinal P-glycoprotein and hepatic and intestinal cytochrome P450 enzymes [13,14]. There was no significant difference in mean skin and whole blood ciclosporin concentrations between dogs given ciclosporin alone (5 mg/kg/day) and dogs that received a combination of ciclosporin and ketoconazole dosed at 2.5 mg/kg/day each for 7 days [13]. In contrast to the ciclosporin brand Atopica’s rough expense of USD 8–9.5/kg for large-weight dogs above 20 kg, generic ketoconazole has an average price of USD 1–1.2 per 200 mg tablet in the United States. However, ketoconazole at a median dosage of 7.8 mg/kg may show adverse effects (e.g., vomiting, inappetence, diarrhea, hepatotoxicity) that may limit the use in some patients [20]. In this study, four PF dogs achieved CR without severe side effects with a combination of ciclosporin (Atopica) and ketoconazole at a mean daily dose of 3 mg/kg and 3.6 mg/kg, respectively. Ciclosporin/ketoconazole drug combination offers another affordable option for PF dogs, for which Atopica standalone therapy may be expensive; however, potential ketoconazole associated side effects need to be carefully evaluated in these patients.

Ciclosporin-induced gastrointestinal side effects have been observed in canine studies, with a prevalence of vomiting, diarrhea, and inappetence at 25%, 15%, and 2%, respectively [14]. In our study, the mild transient adverse effects of oral modified ciclosporin and GCs included gastrointestinal upset (vomiting, diarrhea, and/or inappetence) in six dogs. However, none of the dogs needed to discontinue therapy due to aforementioned side effects. The abnormalities regarding blood work results after three months of treatment (e.g., thrombocytosis, hyperglycemia) in Dog 8 were likely associated with chronic GC use [21,22].

Treatment efficacy of drugs in canine PF has been previously evaluated using clinical scoring groups (e.g., no, partial and complete clinical remission) [10,11], PF extent and severity index [6], and Pemphigus Disease Area Index [23]. However, none of these scoring tools has been validated for canine PF. A validated clinical scoring system for canine PF is essential for prospective randomized controlled studies and comparative clinical trials between different immunomodulators. Our study utilized previously published remission outcomes from studies evaluating the efficacy of oral mycophenolate mofetil and GCs combination therapy in canine PF [4]. In contrast to oral mycophenolate mofetil/GCs therapy, which induced CR in only 2 out of 11 PF dogs [4], oral ciclosporin with GCs achieved CR in 9 out of 11 dogs in this study.

To the best of the author’s knowledge, it is unknown whether previous administration of immunosuppressive dosages of GCs or other drugs could potentially induce treatment-refractory PF in dogs. In this study, eight PF dogs (Dog 2, 3, 5, 6, 7, 8, and 9) received previous immunosuppressive dosages of GCs or other immunosuppressive medications before oral ciclosporin was initiated; seven of these eight PF dogs achieved CR during the induction phase.

## 5. Conclusions

In conclusion, this study provides additional insights into the clinical utilization of oral modified ciclosporin as an adjunct immunosuppressive agent for achieving and maintaining CR in dogs with PF. The main limitations of our study include the retrospective nature, small sample size, and lack of a control group. Further prospective comparative efficacy studies of oral modified ciclosporin and other immunosuppressants for the canine PF treatment should be performed. However, a validated clinical score should be first established so that the future comparative studies can be appropriately evaluated for their success.

## Figures and Tables

**Figure 1 vetsci-09-00153-f001:**
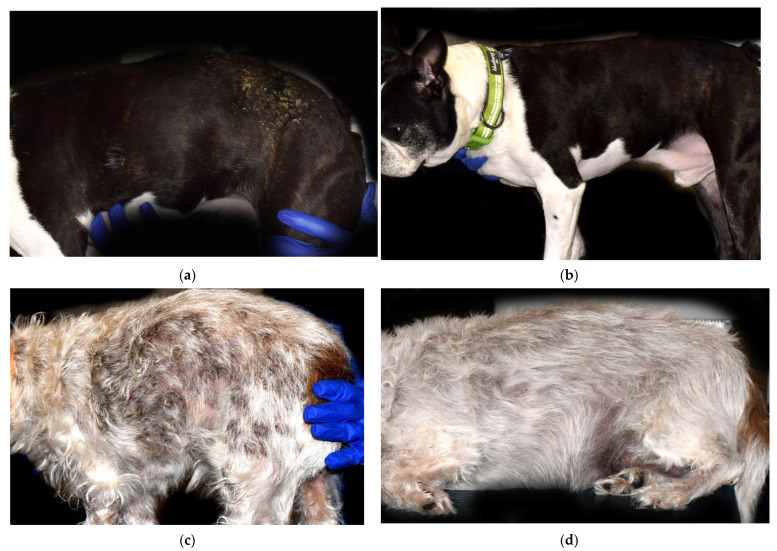
Clinical presentation of two dogs with PF before and after treatment with combined GC and modified oral ciclosporin (Atopica; Elanco, Greenfield, IN, USA). (**a**,**b**) Dog 2 on Day 0 (**a**) and on Day 56 (**b**) after Atopica and GC administration, in complete remission (CR). On Day 0 (**a**), diffuse hypotrichosis and moderate erythema over the lateral thorax and hip with multifocal areas of crusting. On Day 56 (**b**), multifocal areas of hypotrichosis with no active crusts. Dog 4 on Day 0 (**c**), and on Day 57 (**d**), in complete remission (CR). On Day 0 (**c**), large area of multifocal to coalescing crusting over the dorsolateral trunk and hip areas. On Day 57 (**d**), no active skin lesions of canine PF visible.

**Table 1 vetsci-09-00153-t001:** Dogs treated with oral modified ciclosporin (CSA): dosage, duration, concomitant glucocorticoid (GC) and ketoconazole (KC) doses, clinical outcome, and reported adverse effect. GC, glucocorticoids; KC, ketoconazole; Mometasone furoate 0.1% topical, Mometasone; CR, complete remission; POR, poor response; NRR, not reached remission.

Dogs	Breed and Lesion Distribution	CSA Initial Dose (mg/kg/day)	CSA Treatment (days)	Concomitant GCs/KC	GC/KC Dose (mg/kg/day)	Remission Outcome	Time to Achieve CR/PR	Adverse Effects
1	Shih Tzu, facial and truncal	7.5	1510	Prednisolone, Mometasone	1.5	CR	119	None
2	Mixed breed, facial and truncal	6.4	534	Prednisolone	1.9	CR	56	None
3	Labrador retriever, facial and truncal	2.2	144	Prednisone	1.1	CR	24	Diarrhea
4	Boston terrier, truncal	9.6	1081	Prednisone, Mometasone	2.5	CR	57	Inappetence
5	Maltese, facial and truncal	5.2	413	Prednisone	2	CR	32	Diarrhea
6	Labrador retriever, facial and truncal	3.5	344	Prednisone and Mometasone Ketoconazole	1.7 5.2	CR	62	None
7	Beagle, facial and truncal	2.9	597	Prednisone, Ketoconazole	1.1 2.9	CR	53	Vomiting
8	Labrador retriever, facial and truncal	3	155	Prednisone, Mometasone Ketoconazole	0.84 4	CR	89	None
9	Mixed breed, facial and truncal	2.4	275	Dexamethasone, Mometasone Ketoconazole	0.1 2.4	CR	89	Diarrhea
10	Maltese, facial and truncal	5	159	Prednisone, then switched to Dexamethasone	1 0.1	POR	NRR	None
11	English bulldog, facial and truncal	5.5	337	Prednisone	1.7	POR	NRR	Inappetence

## Data Availability

Data sharing not applicable. No new data were created or analyzed in this study.

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
