# Peer review of "A Retrospective Evaluation of the Steroid-Sparing Effect of Oral Modified Ciclosporin for Treatment of Canine Pemphigus Foliaceus"

_vetsci, 2022, doi:10.3390/vetsci9040153_

Round 1

Reviewer 1 Report

The manuscript submitted for review is actually a copy of the work published in Veterinary Dermatology; 33:77-e26, 2022. The only difference is in the drug analyzed. In the above-mentioned study, mycophenolate mofetil (MMF) was assessed, while in the study submitted for review it is ciclosporin. The layout of the work, the subtitles, and even the content of the introduction and the results are described in almost the same way. The authors did not even try to change the title of the work. In the description of Table 1, there are the abbreviations PU/PD which are not included in that table. Such description can be found in the table of work published in Veterinary Dermatology – was copy-paste used?

Retrospective papers are very important, but should describe the research accurately, moreover, the purpose of such research is to clearly present the data to the reader so that the reader does not have to search for data in the quoted publication every time. In this manuscript, the authors very often used a short description, which in my opinion should be described in more detail (eg: "All dogs diagnosed with PF and treated with orally modified cyclosporin were included in the study as described previously [6]").

I am not a supporter of duplicating works, so I do not recommend publishing this manuscript in the form submitted for review. In fact, the authors should have made one good publication of these two manuscripts, rather than splitting them into two nearly identical copies.

Moreover, this work bears signs of autoplagiarism.

Author Response

Dear reviewer, we have answered all the comments in the attached document

Reviewer 2 Report

This is a nicely written paper that provides useful information for practitioners treating this disease.  As with all retrospective studies, the information has to be interpreted with caution, but I believe that is a matter for the readers, not the authors.  I recommend publication.

Author Response

Thank you for your comments

Reviewer 3 Report

This is a straight-forward retrospective study on pemphigus and its treatment with cyclosporine and glucocorticoid.

The presentation is very clear and easy to follow.

The only concern is that it might be more discouraged to use ketoconazole solely as purpose to reduce cyclosporine dose. As mentioned, use of ketoconazole can be associated with side effects. This should be stressed a bit stronger.

Minor:

Line 181 required occasional intermittent oral and topical GCs to maintain PF. This is misleading, would sugeest to replace by: required occasional intermittent oral and topical GCs to maintain success of PF treatment?

Author Response

Thank you for your comments. we have addressed all the comments.

Round 2

Reviewer 1 Report

I stand by my decision to reject this manuscript. If the Authors and the Editor do not agree with my opinion, please appoint a different reviewer.